# User Segmentation in Recommender Systems: Problem Formulation, Algorithms, and Evaluations

## Abstract

Personalized recommendations are key for recommendation systems, which make profits by tailoring marketing strategies and targeting product offerings to distinct user groups. However, current personalized recommendation algorithms do not consider structures in user groups thereby producing non-robust recommendations. Our key insight is to integrate the previously ignored local structure of users into the recommendation algorithm, where we perform a structured user segmentation that considers the hierarchy in user structure. This improves the quality of the recommendations since they enhance their capability to decipher the essence of the user preferences over the noise. Capitalizing on the inherent hierarchical structure of user segments, our method reduces the model size and results in improved accuracy. We conduct experiments using four different approaches for computing user segments and evaluate their performance across various hyperparameter configurations. As far as we know, this is the first comprehensive evaluation on understanding the right user structure to employ for recommendations. Our results demonstrate that our method yields significant improvements in performance metrics across three diverse datasets. The improvement ranges from 9% to 13% across 5 well-known metrics for a large-scale dataset and around 1-2% on two other small datasets. It also gives a significant improvement of around 20% for Relevance to Other Users (ROU) metric that captures the proportion of similar users who have liked an item being recommended to the user.

## 1 Introduction

User segmentation and market segmentation are methodologies employed to categorize and comprehend distinct groups within a target audience or market, with the aim of refining strategies and offerings. User segmentation entails dividing a user base into sub-groups based on a variety of attributes, such as behavior, demographics, usage patterns, or preferences (Cooil et al., 2008). In contrast, market segmentation involves partitioning a broader market into sub-groups of consumers who share similar characteristics, needs, or behaviors, potentially necessitating separate products or marketing strategies (Yankelovich & Meer, 2006; Wind & Cardozo, 1974).

The increasing use of recommender systems in various fields has led to a higher demand for personalized recommendations (Zhang et al., 2019). Grouping users based on behavior, preferences, or demographics is now a key strategy for businesses (Su & Khoshgoftaar, 2009). This approach allows for tailored customization of products and services, leading to more effective marketing, increased user engagement, and higher profits. It also helps in efficient resource use and understanding of underserved segments, improving investment returns. Segmenting users provides insights for developing products that cater to specific needs, increasing user satisfaction and loyalty (Tynan & Drayton, 1987). It helps in maximizing Customer Lifetime Value (CLV) by providing personalized experiences (Vanderveld et al., 2016). This strategy also improves the effectiveness of advertising and communication. Effective user segmentation leads to a cycle of continuous improvement. It provides valuable data that can refine strategies, enhance understanding of user needs, and contribute to business growth.

This problem is hard to solve because it is difficult to derive a good representation of user characteristics form the rating matrix directly. There has been work done in the past that adds a segmentation component to the recommender system such as Yang et al. (2020) and Sun et al. (2021) that use the segments for information sharing and a source of metadata. The recommendations provided by them are different for each user as in the case of a conventional personalized recommendation algorithm but with an additional step that involve user segments. Although this reduces the noise to a certain extent, it also makes the method computationally more expensive. Our method, not only reduces the influence of noise in recommendations but also the model size of the recommendation system.

We leverage the concept of Matrix Factorization (Mnih & Salakhutdinov, 2007) that gives a user latent vector matrix that represents the characteristics of each user very well. This representation is used for user segmentation and solves the problem easily. Once we obtain the segments we provide the same recommendations for all the users belonging to the same segment.

The key insight of the work is that the quality of a recommendation can be improved if the similarities between users and their characteristics are also given importance. Also, this work shows that although each user is different, good quality recommendations may also be tailored for a group of similar users. This makes the process comparitively simpler as opposed to doing an overly fine-grained analysis to obtain personalized recommendations for each user which are over sensitive to noise and outliers.

Our results demonstrate that our method yields significant improvements in performance metrics across three diverse datasets. The improvement ranges from 9% to 13% across 5 well-known metrics for a large-scale dataset and around 1-2% on two other small datasets. It also gives a significant improvement of around 20% for Relevance to Other Users (ROU) metric that captures the proportion of similar users who have liked an item being recommended to the user.

The contributions of this paper are summarized as follows:

- **Establishment of Evaluation Framework and Dataset Creation:** We craft three datasets derived from existing ones to address this problem domain. We also present a clear definition of recommendation relevance for users. Additionally, we define a methodology for assessing user segmentation models applicable to recommender systems.

- **Novel Approach to User Segmentation in Recommender Systems:** We introduce a novel method for recommending the same items to groups of similar users, a novel concept in recommender system research.

- **Provision of Robust Baseline Methods for Future Research:** We present four distinct algorithms for achieving user segmentation and present their outcomes as robust baseline approaches. These algorithms serve as valuable reference points for future research endeavors aimed at enhancing user segmentation models for recommender systems.

## 2  Methodology

**Problem.** We have a rating matrix $R \in \mathbb{R}^{I \times J}$, where $I$ denotes the number of users and $J$ denotes the number of items. We want to group these $I$ users into $S$ segments (clusters) where the users that belong to the same segment have similar interests or taste. Using this, we provide the same item recommendations for all users in the same segment. This problem can be formulated as the method of finding the optimal $U^{meta} \in \mathbb{R}^{S \times K}$, where $S$ is the number of user segments and $K$ is the size of a vector that describes each user segment. $u_s^{meta}$ at index $s$ denotes the vector consisting of the characteristics of user segment $s \in S$ that can be mapped to multiple similar users that belong to $s$. This can be used to obtain item recommendations for all the users associated with the segment $s$. In a recommendation system without user segmentation, we have $S = I$, i.e., each segment has only one user.

Firstly, we introduce our prediction model. Then we go on to introduce our approach and the optimization problem that we are going to solve. We start by outlining collaborative deep learning (CDL) (Wang et al., 2015), which serves as our base model. Next, we introduce the user segmentation component built upon

CDL, creating a new training objective. Finally we elaborate on how prediction is done for the models and how we evaluate our method.

## 2.1 Prediction

In CDL (Wang et al., 2015), the predicted rating is computed as:

$$R_{ij} = u_i^T v_j \tag{1}$$

For our method, we segment the test dataset by performing multiple iterations of the segmentation step into test meta users and compute:

$$R_{ij}^{meta} = u_i^{meta\,T} v_j \tag{2}$$

This procedure yields predicted ratings for each meta user in the test dataset. We then extend the recommended items for the meta user to all users within that specific meta user group. Consequently, users within the same segment receive identical item recommendations. To evaluate both K-Means and Hierarchical K-Means variants, we perform K-Means clustering on the test dataset.

It might be contended, that it is not necessary to alter the objective function for CDL to accommodate segmentation. Instead, one could simply apply the segmentation algorithm to the trained model i.e. U obtained after training by the CDL process and compare it against the test data, which has also been segmented into the same user groups. This approach serves as our baseline, and we subsequently demonstrate in our results that training in conjunction with the modified objective indeed results in enhanced recommender system performance. Algorithm 1 describes the inference process in detail.

---

**Algorithm 1** Inference using K-means based segmentation as an example

---

(1) **procedure** INFERENCE(test dataset, matrix factorization model)
(2)     The matrix factorization model consists of the following:
   1. U: user latent vector matrix
   2. $U^{meta}$: user latent vector matrix of kmeans centroids when applied on U
   3. V: item latent vector matrix
(3)     segmented test users $t_u \leftarrow$ cluster centers on performing KMeans clustering on test dataset
(4)     user labels $l \leftarrow$ array($l_i : \forall u_i \in U | u_i$ is a row vector in U where $l_i$ is the user cluster for $u_i$ on performing KMeans clustering on the rows of U)
(5)     item labels $m \leftarrow$ cluster centers on performing KMeans clustering on the transpose of test dataset
(6)     estimation $e \leftarrow U^{meta} V^T$
(7)     recommended items $r \leftarrow e[l]$
(8) **end procedure**

---

## 2.2 Preliminary: Collaborative Deep Learning (CDL)

CDL (Wang et al., 2015), as mentioned previously, is a hybrid recommender system that combines the power of deep learning with collaborative filtering. CDL integrates a Stacked Denoising Autoencoder (SDAE) to capture content-based insights and Matrix Factorization (MF) for collaborative filtering, which captures the unique characteristic of an individual user and recommends similar products to a user with similar attribute. Once, these characteristics of each individual user is captured, a segmentation step can be added to group the users and thus derive their salient characteristics and relationships reducing the sensitivity towards noise in recommendations.

---

**Algorithm 2** Collaborative Deep Learning Without User Segmentation

---

(1)  **procedure** TRAINING(content dataset, ratings dataset, $Ep, \lambda_u, \lambda_v, \lambda_w, \lambda_n$)
(2)      $n \leftarrow -1$
(3)      **while** $n \neq Ep - 1$ and convergence in kmeans clustering<threshold **do**
(4)          $n \leftarrow n + 1$
(5)          **for** $i \in [0...u]$ **do**                                              ▷ u is the number of users
(6)              $u_i \leftarrow (VC_iC^T + \lambda_u I_K)^{-1}VC_iR_i$              ▷ R from ratings dataset & C is a diagonal matrix
(7)              **for** $j \in [0...v]$ **do**                                       ▷ v is the number of items
(8)                  $v_j \leftarrow (UC_iU^T + \lambda_v I_K)^{-1}(UC_jR_j + \lambda_v f_e(X_{o,j*}, W^+)^T)$
(9)              **end for**
(10)         **end for**
(11)         Loss function L computed by Equations 3 to 7
(12)         Train collaborative deep learning autoencoder using loss L, content dataset and updated V
(13)     **end while**
(14)     Encode the content dataset using the autoencoder
(15)     Optimize U and V completely holding the SDAE latent layer fixed
(16) **end procedure**

---

It solves the following optimization problem by maximizing the joint log-likelihood of $U$, $V$, $\{X_l\}$, $X_c$, $\{W_l\}$, $\{b_l\}$, and $R$ given $\lambda_u$, $\lambda_v$, $\lambda_w$ and $\lambda_n$:

$$L = -\frac{\lambda_u}{2}\sum_i \|u_i\|_2^2 \tag{3}$$

$$-\frac{\lambda_w}{2}\sum_l (\|W_l\|_F^2 + \|b_l\|_2^2) \tag{4}$$

$$-\frac{\lambda_v}{2}\sum_j \|v_j - f_e(X_{0,j*}, W^+)^T\|_2^2 \tag{5}$$

$$-\frac{\lambda_n}{2}\sum_j \|f_r(_{0,j*}, W^+) - X_{c,j*}\|_2^2 \tag{6}$$

$$-\sum_{i,j} \frac{C_{ij}}{2}(R_{ij} - u_i^T v_j)^2 \tag{7}$$

Each term has the following intuition with $\lambda_u$, $\lambda_v$, $\lambda_w$ and $\lambda_n$ as hyperparameters:

- **Regularization for user latent vectors (Equations 3 and 4)** U obtained from the matrix factorization process where $u_i$ is a latent user vector of user i, $\{W_l\}$ is the weight matrix of the SDAE at layer l and $\{b_l\}$ is its bias

- **Encoding Loss in SDAE (Equation 5)** $v_j$ is a latent item vector of item j. $f_e(X_{0,j*}, W^+)$ is the encoder function that takes content vector $X_{0,j*}$ as input and computes the encoding of the item given. $W^+$ is the collection of all layers of weight matrices and biases.

- **Reconstruction Loss in SDAE (Equation 6)** $f_r(_{0,j*}, W^+)$ takes the same input as the encoder function and computes the encoding and then the reconstruction vector that is compared against the output of the final layer of the encoder given by $X_{c,j*}$ where c is used to indicate the clean input.

- **Error of Predicted Ratings from the Matrix Factorization Model (Equation 7)** $R_ij$ is the rating for item j given by user i and $C_ij$ is a confidence parameter that is based on $R_{ij}$.

On computing the gradients, for the matrix factorization optimizer, the update steps for U and V are as follows:

$$u_i = (VC_iV^T + \lambda_u I_K)^{-1}(VC_iR_i) \tag{8}$$

$$v_i = (UC_iU^T + \lambda_v I_K)^{-1}(UC_jR_j + \lambda_v f_e(X_{0,j*}, W^+)^T) \tag{9}$$

- Here

    - $U = (u_i)_{i=1}^I$
    - $V = (v_j)_{j=1}^J$
    - $C_i = diag(C_{i1}, C_{i2}, ..., C_{iJ})$
    - $R_i = (R_{i1}, R_{i2}, ..., R_{iJ})^T$

    is the column vector containing all the ratings of user i.

- $I_K$ is an identity matrix of size K which is the latent vector size on performing probabilistic matrix factorization.

The CDL algorithm is given by Algorithm 2.

## 2.3 Our Approach: The Segmentation Step

---
**Algorithm 3** Collaborative Deep Learning With User Segmentation
---
(1) **procedure** TRAINING(content dataset, ratings dataset, $Ep, \lambda_u, \lambda_v, \lambda_{meta}, \lambda_w, \lambda_n$)
(2)     $n \leftarrow -1$
(3)     previous user labels $p \leftarrow$ integer array$[-1, -1, \ldots, -1]_u$
(4)     user labels $l \leftarrow$ integer array$[-2, -2, \ldots, -2]_u$
(5)     Randomly initialize $U^{meta}$
(6)     **while** $n \neq Ep - 1$ and convergence $<$ threshold **do**
(7)         $n \leftarrow n + 1$
(8)         $p \leftarrow l$
(9)         Compute $U^{meta}$ and $l$ by the Segmentation Step in Sec. 2.3
(10)         **for** $i \in [0, \ldots, u]$ **do**
(11)             $u_i \leftarrow (VC_iC^T + \lambda_u I_K)^{-1}(VC_iR_i + \lambda_{meta}u_{s|i \in s}^{meta})$
(12)             **for** $j \in [0...v]$ **do**
(13)                 $v_j \leftarrow (UC_iU^T + \lambda_v I_K)^{-1}(UC_jR_j + \lambda_v f_e(X_{o,j*}, W^+)^T)$
(14)             **end for**
(15)         **end for**
(16)         Loss function L computed by Equation 11
(17)         Train collaborative deep learning autoencoder using loss L, content dataset and updated V
(18)     **end while**
(19)     **while** convergence $<$ threshold **do**
(20)         $p \leftarrow l$
(21)         Compute $U^{meta}$ and $l$ by Segmentation Step
(22)     **end while**
(23)     Encode the content dataset using the autoencoder
(24)     Optimize U and V completely holding the SDAE latent layer fixed
(25) **end procedure**
---

In the CDL (Wang et al., 2015) process, during each training cycle, the SDAE adjusts the hidden representations of items, which are then passed to the matrix factorization optimizer. This optimizer updates both user and item representations to include the knowledge gained from the item representations adjusted by the SDAE. Subsequently, the modified item representations are used to update the SDAE's internal weights, allowing it to learn from the matrix factorization model. However, directly performing the optimization like this can cause overfitting which makes the model very senstive to noise if the data is noisy. It also ignores the relationships that these users may have with each other and the possibility of a heirarchical structure in their similarities.

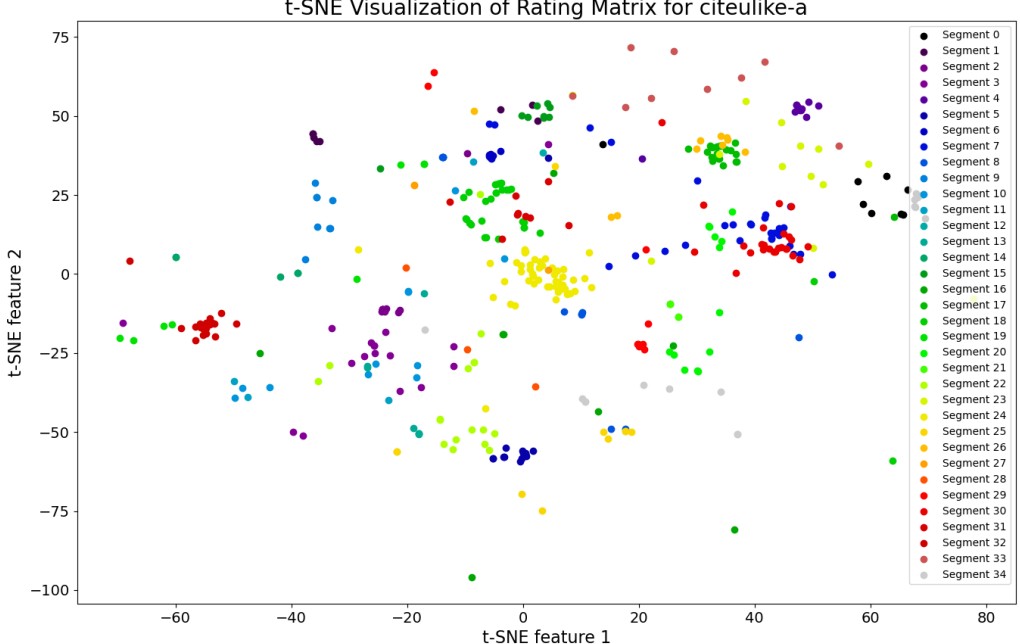

Figure 1: The ratings of users belonging to 35 segments are shown and it is seen that users belonging to the same segment cluster together.

We propose to tackle the problem by adding additional structure to the user groups. Our key insight is that the representation of the user group must be included in the optimization process to make the recommendation robust to individual outliers. The fact that user ratings exist in clusters can be seen in Figure 1 which shows the T-SNE plot of all the users that belong to 35 segments. Before the matrix factorization optimization, the latent user vectors are divided into multiple segments and each segment is represented by a vector that we call the meta user vector. This vector is used to optimize the matrix factorization model by changing the updation of U to:

$$u_i = (VC_iV^T + (\lambda_u + \lambda_{meta})I_K)^{-1}(VC_iR_i + \lambda_{meta}u_{s|i \in s}^{meta}) \quad (10)$$

Here $U^{meta}$ is a matrix where each row $u_{s|i \in s}^{meta}$ is the mean of the latent user vectors that belong to segment i. We call $u_s^{meta}$ the meta user representation. $\lambda_{meta}$ is an additional hyperparameter. The CDL training along with the segmentation step starts only after one epoch without the segmentation step so that we get the initial U matrix. Thus the equation for the loss modifies to become:

$$L \leftarrow \frac{1}{2}\sum_i(\lambda_u||u_i||_2^2 + \lambda_{meta}||u_i - u_{s|i \in s}^{meta}||_2^2) - \frac{\lambda_w}{2}\sum_l(||W_l||_F^2 + ||b_l||_2^2) - \frac{\lambda_v}{2}\sum_j||v_j - f_e(X_{o,j*}, W^+)^T||_2^2 -$$

$$\frac{\lambda_n}{2}\sum_j||f_r(X_{o,j*}, W^+) - X_{c,j*}||_2^2 - \sum_{i,j}\frac{C_{ij}}{2}(R_{ij} - u_i^Tv_j)^2 \quad (11)$$

We explore different techniques of obtaining $U^{meta}$, that are described below.

### 2.3.1  Our Approach: K-Means Clustering

In this method, the segmentation step is one iteration of K-Means clustering (Hartigan & Wong, 1979). This iteration consists of:

- Assigning every user representation in U to the nearest meta user i.e. the nearest $u_s^{meta}$ by Euclidean distance, when $U^{meta}$ is given.

- Recomputing the means for newly assigned latent user representations grouping them by the meta user they belong to, to obtain a new $U^{meta}$ value.

$U^{meta}$ is initialized to be S random set of user latent vectors U, where S is also the number of user segments. The algorithm for the complete process of training using K-Means Clustering as the segmentation step is given in Algorithm 4.

### 2.3.2 Our Approach: Hierarchical K-Means Clustering

K-Means clustering requires to specify the number clusters at the beginning of the algorithm. However, once the number is chosen, they are fixed and cannot be adapted. We further tackle this problem adapting a hierarchical k-means clustering (Murtagh & Contreras, 2012), which can adapt the number of clusters as algorithm runs. We start with an number of clusters that is over complete. After recalculating cluster centers, we merge n pairs of the closest clusters. In our case a total of two pairs at each iteration (n=2). Then, we compute new cluster centers and reassign each user's meta user based on these merged cluster centers. Initially, the set of meta users, $U^{meta}$, is the same as the set of users U. However, as we progress through iterations, the size of $U^{meta}$ gradually decreases until it reaches a value of S. In this method, the number of user segments are not enforced on the training process at the very beginning but it is gradually adapted, allowing our algorithm to be more flexible and generalized.

### 2.3.3 Other Segmentation Methods

The segmentation step can be any method that group users. Gaussian Mixture Models (GMM) can also serve as a segmentation step, similar to K-Means Clustering. Instead of being one iteration of K-Means clustering, it can be one iteration of GMM clustering.

Another prevalent method for grouping related entities is Community Detection. In this case, the segmentation step involves a graph algorithm performing Community Detection, with each community representing a user segment.

Appendix F.1 describes Community Detection and Appendix B describes GMMs used as the Segmentation Step.

### 2.4 Evaluation Scheme

The following metrics are used to evaluate the recommendations for the top M items recommended to the users:

- **Recall:**
$$\text{recall@}M = \frac{\text{number of relevant among the top } M \text{ for the user}}{\text{total number of relevant items for the user}} \tag{12}$$

- **Precision:**
$$\text{precision@}M = \frac{\text{number of relevant among the top } M \text{ for the user}}{M} \tag{13}$$

- **Mean Reciprocal Rank (MRR):** The reciprocal of a rank is the multiplicative inverse of the rank of a relevant item recommender. Hence, MRR is defined as

$$\text{MRR@}M = \sum_{i=1}^{M} \frac{\mathbb{1}_{item_i \text{ relevant for user}}}{i} \tag{14}$$

For our experiments we check for MRR@M and the reciprocal rank for the first relevant item(RR@1) to see at what rank the first relevant item for the user is recommended.

- **Normalized Discounted Cumulative Gain (nDCG):** nDCG measures the effectiveness of a ranking by considering both the relevance of the items and their position in the list. Discounted Cumulative Gain (DCG) weighs each relevance score based on its position and sums it up as

$$\text{DCG@}M = \sum_{i=1}^{M} \frac{\mathbb{1}_{item_i \text{ relevant for user}}}{log_2(i+1)} \tag{15}$$

This score depends on the number of items recommended to the user and makes it difficult to compare two recommenders that recommend a different number of items. That is why we use nDCG defined as:

$$\text{nDCG@}M = \frac{\text{DCG@}M}{\text{IDCG@}M} \tag{16}$$

where IDCG@$M$ is the score when we recommend all of the most relevant items first. In our experiments we give equal importance to all relevant items, so the ordering for IDCG doesn't really matter.

- **Relevance to Other Users (ROU):** We introduce a novel metric, specifically tailored to the User Segmentation problem, to assess the impact of user collaboration and the influence of similar users on recommendation generation. This metric, termed "Relevance to Other Users" (ROU), is defined as follows

$$\text{ROU@}M = \sum_{i=1}^{M} \mathbb{1}_{item_i \text{ not rated by user}} \frac{\text{number of other users in the segment who have rated } item_i}{\text{number of users in segment} \times i} \tag{17}$$

This metric can be conceptualized as the aggregate of the product of two factors: the proportion of other users who have rated an item new to the user under consideration and the reciprocal rank of that item for the user under consideration. The first part provides an indication of the item's relevance for the user, while the reciprocal rank reflects the importance assigned to that item during the inference process for the user.

- **Relevance of an Item:** In determining item relevance, we assess whether it bears similarity to items that a user has positively rated. To identify item similarities, we employ K-Means clustering across the entire item dataset. Items falling within the same cluster are regarded as similar. Consequently, when a user is recommended an item belonging to the same cluster as any item they have rated positively, that recommendation is deemed relevant.

These metrics are computed and compared for both the baseline and our method for every variant. The percentage increase or decrease for every metric of our method with respect to the baseline is analyzed.

## 3 Experiments

### 3.1 Datasets

We evaluate our model using three diverse datasets of varying sizes. Two of these datasets, namely, *citeulike-a* and *citeulike-t*, were previously utilized in the study conducted by Wang et al. (2015). *citeulike-a* and *citeulike-t* are created from CiteULike a platform that allows users to create their own collection of articles forming a citation dataset. The task is to recommend articles to users based on the articles for which they have rated. For each article, the embeddings used by the SDAE consists of the title of the paper and its abstract. *citeulike-a* comprises 5,551 users and 16,980 items, while *citeulike-t* encompasses 7,947 users and 25,975 items. For these two datasets, for each user 10 items that they rated were used for training.

Additionally, we incorporate a third dataset from Amazon, which combines item descriptions and user ratings for products rated by the users from 14 different countries. This results in a dataset featuring 26,165 users and 31,245 items. The content-based information for this dataset is derived from product titles and descriptions. The recommendation task is to suggest to the users products that they may be interested in

Table 1: Metric values for K-Means

| Experimental Setting | Recall@20 | Precision@20 | nDCG@20 | MRR@20 | RR@1 |
|---|---|---|---|---|---|
| **citeulike-a:** | | | | | |
| Proposed Model | 0.0284 | 0.4980 | 0.6745 | 1.9494 | 0.6337 |
| Baseline Model | 0.0283 | 0.4899 | 0.6714 | 1.9262 | 0.6303 |
| Change(%) | +0.35 | +1.65 | +0.46 | +1.20 | +0.54 |
| **citeulike-t:** | | | | | |
| Proposed Model | 0.0088 | 0.1725 | 0.2299 | 0.6706 | 0.2176 |
| Baseline Model | 0.0089 | 0.1664 | 0.2242 | 0.6476 | 0.2113 |
| Change(%) | -1.12 | +3.66 | +2.59 | +3.55 | +2.98 |
| **Amazon:** | | | | | |
| Proposed Model | 0.0105 | 0.1011 | 0.2067 | 0.4381 | 0.1851 |
| Baseline Model | 0.0096 | 0.0909 | 0.1947 | 0.3857 | 0.1672 |
| Change(%) | +9.37 | +11.22 | +6.16 | +13.58 | +10.70 |

based on the products that they have rated. For the training process in this dataset, we considered a much stricter setting where for each user only one of the items that they had rated was given.

We consider user-item interactions as positive ratings if a user has rated an item, and conversely, as non-ratings if no rating has been provided when computing the relevance of recommended items.

### 3.2 Experimental Setting

For every dataset, we compare we compare our model against the baseline model. We do this for multiple number of user segment configurations. The process we employ to generate each datapoint involves conducting a series of experiments with various hyperparameter settings(for the $\lambda$ hyperparameters). For each type of segmentation step, we identify and select the hyperparameter configuration that yields the most favorable outcomes across all configurations of number of user segments considered. Therefore, while each data point is derived from numerous experiments, it effectively encapsulates the findings from the single most optimal trial for the best possible same set of $\lambda$ hyperparameters.

For our evaluation, we follow the definition of relevance of an item given in Section 2.4 but consider an item to be relevant only if the user has not rated it for training. More information on the Evaluation Settings can be found in Appendix C. Appendix D and Appendix E also describes how our method performs if the items already rated by the user during training are considered as relevant.

## 4 Results

### 4.1 K-Means as a Consistent Segmentation Step

Table 1 illustrates that K-Means Clustering, employed as the segmentation step, consistently surpasses the baseline performance across various datasets. Although the enhancements for the *citeulike-a* is moderate, there is more pronounced improvement in the *citeulike-t* and amazon datasets, particularly for Amazon. However, an examination of metric values, particularly the MRR and RR metrics, for the *citeulike-a* dataset reveals significantly improved values. These metric values demonstrate competitiveness with recommender systems that do not incorporate user segmentation. This is crucial because users are primarily concerned with the quality of recommendations rather than the internal workings or optimizations of the model.

The reciprocal rank (RR) values indicate that the first relevant item typically appears around the second or third position for *citeulike-a*, around the fifth position for *citeulike-t* and lastly around the tenth position for amazon.

Table 2: Metric values for Heirarchical K-Means

| Experimental Setting | Recall@20 | Precision@20 | nDCG@20 | MRR@20 | RR@1 |
|---|---|---|---|---|---|
| **citeulike-a:** 50% size | | | | | |
| Proposed Model | 0.0292 | 0.4999 | 0.7017 | 2.0186 | 0.6711 |
| Baseline Model | 0.0288 | 0.4892 | 0.6997 | 1.9850 | 0.6619 |
| Change(%) | +1.38 | +2.19 | +0.28 | +1.69 | +1.39 |
| **citeulike-t:** 25% size | | | | | |
| Proposed Model | 0.0090 | 0.1734 | 0.2374 | 0.6849 | 0.2268 |
| Baseline Model | 0.0090 | 0.1684 | 0.2350 | 0.6686 | 0.2243 |
| Change(%) | 0.00 | +2.97 | +1.02 | +2.43 | +1.11 |
| **Amazon:** 20% size | | | | | |
| Proposed Model | 0.0104 | 0.0996 | 0.2052 | 0.4310 | 0.1813 |
| Baseline Model | 0.0097 | 0.0911 | 0.2014 | 0.4027 | 0.1768 |
| Change(%) | +7.22 | +9.33 | +1.89 | +7.03 | +2.60 |

All of these results are collected when the number of user segments reduces the model size to approximately 10% of its original size. Achieving these good results with a significant reduction in model size in one of the main contributions of our work.

## 4.2 Heirarchical K-Means as a Useful Segmentation Step

Table 2 illustrates the utility of Hierarchical K-Means as a segmentation step. While the optimal outcomes may not occur with a model size reduction as small as the 10% of K-Means, the displayed results suggest a significant improvement in metric values. This shows that there is some use of including the heirarchical structure of the users into the model. This may be due to the inherent characteristic of Hierarchical K-Means which makes it gradually reduce the number of segments in the training process to S. The number of epochs it undergoes depends on the rate at which it converges to S segments and so it helps in making sure that the model doesn't overfit for a given number of segments.

Appendix D illustrates the model's performance across various configurations of user segmentation. And, Appendix E illustrates how the performance of our method evolves as we progressively refine the evaluation by increasing the number of relevant item clusters. This adjustment results in a reduction of elements allocated to each relevant cluster, intensifying the evaluation criteria, as the pool of items deemed relevant for a user diminishes with the rising count of relevant item clusters.

## 4.3 The Relevance of Recommendations for Users Based on Items Trained by Similar Users

From Table 3, it can be seen that in most cases for the CiteULike datasets the Proposed Model does better than the Baseline in terms of providing new item recommendations to a user which has been liked by a similar user.

For the Amazon dataset, the results exhibit extremes that escalate in magnitude with an increase in the number of user segments. The occurrence of $+\infty$ when the number of segments is half the number of users is attributed to the fact that the baseline ROU@20 for all models is 0, and any improvement by KMeans and Hierarchical KMeans over this baseline is infinitely more. And the community detection results are not applicable since community detection doesn't work for the amazon dataset as descibed in Appendix F.2.

## 4.4 Qualitative Analysis

For qualitative analysis, we take an illustrative segment from the proposed model using K-Means as the segmentation step on the *citeulike-a* dataset. Appendix G shows an example of a cluster of users and the items that they have rated. It also shows the items recommended to the users of the segment for our method

Table 3: The Relevance of Recommendations for Users Based on Items Trained by Similar Users (ROU@20)

| # User Segments | Method | % Change in Metric from Baseline | | |
| | | citeulike-a | citeulike-t | amazon |
| --- | --- | --- | --- | --- |
| | KMeans | +8.82 | +20.67 | +7.26 |
| 10% of #users | HK-Means | -0.43 | +13.25 | -16.49 |
| | KMeans | +13.23 | +23.99 | -70.08 |
| 20% of #users | HK-Means | +11.16 | +21.19 | -9.55 |
| | KMeans | +11.52 | +21.03 | +43.15 |
| 25% of #users | HK-Means | +11.37 | +35.31 | +14.69 |
| | KMeans | +17.21 | +28.41 | -90.04 |
| 33.33% of #users | HK-Means | +4.86 | +29.16 | +80.46 |
| | KMeans | +14.29 | +32.46 | $+\infty$ |
| 50% of #users | HK-Means | +14.85 | -41.94 | $+\infty$ |

and the baseline. In Table 11, we present the items that each user within this segment has previously liked and used for training. Upon reviewing the titles in Table 11, it becomes evident that these users share similar preferences, indicating that the proposed model has effectively grouped them into the same segment.

Table 12 provides insights into the recommendations generated by both the proposed model and the baseline model for the specified experimental setup. Notably, the recommendations from both models align with the titles that the users have rated in the past. It is also important to note that the last two recommendations of the baseline are very related to a peer-to-peer file system which may not be as relevant to the user as the rest of the recommendations whereas the all of the recommendations given by our model are very relevant. Additionally, both the proposed model and the baseline recommend some items that users have already rated, suggesting the model's ability to understand user preferences, albeit with potential concerns of overfitting.

Furthermore, both models suggest items that none of the users have encountered during training, yet these recommendations are relevant. This demonstrates the positive impact of user segmentation on the collaborative filtering process. Moreover, our model appears to excel in generating such recommendations, as indicated by the higher ROU@20 indicated by Table 3, in comparison to the baseline. This implies that the items recommended by our model are both relevant to the user and relatively novel, in contrast to the baseline model.

It is also seen that our method is robust to outliers in the user's preferences. For example, the user ratings on which our method is trained on can contain some items that the user may have given a good rating by mistake but they actually dislike. It is seen that the model is able to detect this and find the user's real preferences. Results for this are seen in H.

## 5 Related Work

While Guo et al. (2020) focuses on Knowledge-Graph based Recommender Systems. It discusses a Connection-Based method, which relies on relationships among different entities in Recommender Systems. The concept involves a Meta-Structure method with an objective function similar to ours which is a common recomender system objective along with constraints to learn user and item embeddings. In our case, the objective function combines the CDL recommender system's objective function with similarity constraints that aid in learning user and item embeddings.

When exploring Meta-Structure methods, a prevalent approach involves identifying similarities among either item representations or user representations. For instance, Yu et al. (2013) delves into item similarity to suggest items akin to a user's interests. Recognizing the importance of recommending the same item to similar users, alternate methods explore user similarity without utilizing a segmentation approach. An example is Zhao et al. (2017). RippleNet (Wang et al., 2018) and AKUPM (Tang et al., 2019), try to refine the user representation and consequently their preferences by propagation based methods.

Existing user segmentation methods in recommender systems often employ user segments for information sharing rather than giving every user in the segment the same recommendation. For example, Yang et al. (2020) leverages user segments to enhance information sharing, while Sun et al. (2021) employs user grouping to establish region-dependent crowd preferences for Point-of-Interest recommendations. Tan et al. (2023) uses user segmentation features along with a universal representation of all users to perform conversion rate prediction. Here, given the user segments, the method tries to learn the appropriate representation for these segments. In our approach we not only learn the representation for each user segment but we also learn which users belong to each segment. Luo et al. (2022) utilizes federated learning to protect user privacy, with user segmentation facilitating model exchanges. Similarly, Shams et al. (2021) and Yu et al. (2014) use clustering to expedite new user preference learning. Li et al. (2019) clusters similar users into communities but retains individual preferences in recommendations, in contrast to our approach, which focuses on providing the same recommendations to similar users and model size reduction.

Hierarchical Matrix Factorization (Sugahara & Okamoto, 2024; Melchiorre et al., 2022) addresses our goal to develop a recommendation system that considers user relationships and mitigates noise. Nevertheless, this approach does not inherently yield distinct user groups or segments that could be directly leveraged for marketing strategies, in contrast to our proposed method. While Hierarchical Matrix Factorization could serve as a foundational comparison point for our research, exploring its application as a baseline constitutes an avenue for future work. This brings in the motivation for user segmentation since these methods do not explicitly group users together like user segmentation.

None of these approaches directly addresses the formulated problem and cannot be directly used as baselines. Consequently, we established our own baseline, which involves applying segmentation after training any recommendation model (CDL for our baseline). It is noteworthy that the user grouping techniques employed in these studies could be viewed as potential solutions to the user segmentation problem, serving as a segmentation step or a related procedure. Besides, our method doesn't involve a Knowledge-Graph.

## 6  Discussion and Conclusion

From these results, it is evident that segmentation methods consistently outperform or match the baseline performance. While there are a few configurations where our method exhibits slightly lower performance than the baseline for certain metrics, these instances are outweighted by some of the outcomes especially that which was observed when the model was reduced to 10% using K-Means as the segmentation step. It is also seen that Heirarchical K-Means clustering can also be a good segmentation step when the model size need not be reduced to such a large extent.

Another noteworthy observation is that the ideal configuration concerning the number of user segments varies significantly depending on the dataset's nature. This is apparent from the disparities in results across datasets and the number of user segments. Recent work Charikar et al. (2023) shows that it seems likely that clustering algorithms with theoretical guarantees could be leveraged which would enable us to know how far we are away from the optimal clustering which, in turn, would enable us to alter the clustering to move closer to the optimal number of segments. Determining the most suitable level of segmentation granularity for a specific dataset goes beyond the scope of this research but any clustering approach which allows the ideal number of clusters to be determined can be leveraged in our work.

Also, if this model is deployed into a real-world setting, it wouldn't need to update itself as users are added as long as the users belong to any of the user segments identified by the model. Re-training of the model will only be required when a new kind of user which does not belong to any segment is introduced. This is very useful.

To conclude, this research introduces an innovative recommender system incorporating a user segmentation component, resulting in a substantial reduction in model size and a noteworthy enhancement in its overall performance.

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

## A Detailed Collaborative Deep Learning with K-Means Clustering Algorithm

Algorithm 4 describes the complete process of training using K-Means Clustering as the segmentation step.

**Algorithm 4** Collaborative Deep Learning With K-Means based segmentation as an example

(1) **procedure** TRAINING(content dataset, ratings dataset, epochs $Ep, \lambda_u, \lambda_v, \lambda_{meta}, \lambda_w, \lambda_n$, threshold $t$)
(2)     convergence in kmeans clustering $c \leftarrow 0 \quad n \leftarrow -1$
(3)     previous user labels $p \leftarrow$ integer array$[-1-1...-1]_u$        $\triangleright$ u is the number of users
(4)     user labels $l \leftarrow$ integer array$[-2-2...-2]_u$
(5)     $U^{meta} \leftarrow U[$S random U vectors$]_S$        $\triangleright$ S is the number of user segments
(6)     **while** $n \neq Ep - 1$ and $c < t$ **do**
(7)         $n \leftarrow n + 1$
(8)         $p \leftarrow l$
(9)         distances $d \leftarrow$ array$[00....0]_{u \times S}$        $\triangleright$ Beginning of KMeans iteration
(10)         **for** $i \in [0...u]$ **do**
(11)             **for** $s \in [0...S]$ **do**
(12)                 $d_{is} \leftarrow ||u_i - u_s^{meta}||_2$
(13)             **end for**
(14)         **end for**
(15)         **for** $i \in [0...u]$ **do**
(16)             $l_i \leftarrow \underset{s}{\operatorname{argmin}} u_i$
(17)         **end for**
(18)         **for** $s \in [0...S]$ **do**
(19)             aggregate $a \leftarrow$ array$[00...0]_K$        $\triangleright$ K is the latent size
(20)             number of points $n_p \leftarrow 0$
(21)             **for** $i \in [0...u]$ **do**
(22)                 **if** $s = l_i$ **then**
(23)                     $a \leftarrow a + u_i$
(24)                     $n_p \leftarrow n_p + 1$
(25)                 **end if**
(26)             **end for**
(27)             $u_s^{meta} \leftarrow \frac{a}{n_p}$
(28)         **end for**        $\triangleright$ End of KMeans iteration
(29)         **for** $i \in [0...u]$ **do**        $\triangleright$ matrix factorization optimization step begins
(30)             $u_i \leftarrow (VC_iC^T + (\lambda_u + \lambda_{meta})I_K)^{-1}(VC_iR_i + \lambda_{meta}u_{s|i \in s}^{meta})$
(31)             **for** $j \in [0...v]$ **do**        $\triangleright$ v is the number of items
(32)                 $v_j \leftarrow (UC_iU^T + \lambda_v I_K)^{-1}(UC_jR_j + \lambda_v f_e(X_{o,j*}, W^+)^T)$
(33)             **end for**
(34)         **end for**
(35)         Loss function L computed by Equation 11
(36)         Train collaborative deep learning autoencoder using loss L, content dataset and updated V
(37)         label count $m \leftarrow 0$        $\triangleright$ Checking for convergence in clustering
(38)         **for** $i \in [0...u]$ **do**
(39)             **if** $l_i = p_i$ **then**
(40)                 $m \leftarrow m + 1$
(41)             **end if**
(42)         **end for**
(43)         $c \leftarrow \frac{m}{u}$
(44)     **end while**
(45)     **while** c < 0.99 **do**
(46)         $p \leftarrow l$
(47)         KMeans iteration of lines 9 to 28
(48)     **end while**
(49)     Encode the content dataset using the autoencoder
(50)     Optimize U and V completely holding the SDAE latent layer fixed
(51) **end procedure**

## B    Gaussian Mixture Models(GMM) Clustering as a Segmentation Step

In the segmentation step, an iteration of GMM clustering (Dempster et al., 1977) is applied which consists of the following steps:

- Computing the means (meta user representation) and covariance matrix of U.

- The K-Means method of assigning every user to its meta user. This is a modified version of the original GMM algorithm where the soft probability for each cluster is not updated. It is a simplified approach to the GMM algorithm.

The means are initialized from the classical K-Means algorithm and the covariance matrix is initialized as the identity matrix.

To accomodate for the covariance matrix, the updation step is modified to the following:

$$u_i = (VC_iV^T + \lambda_u I_K + \lambda_{meta}\Sigma^{-1})^{-1}(VC_iR_i + \lambda_{meta}\Sigma^{-1}u_{s|i\in s}^{meta}) \tag{18}$$

Results for various configurations of GMM as a segmentation step are given in Appendices  D and  E.

The limitations of GMM clustering as a segmentation step stem from the fact that after a few iterations, multiple segments merge into a single large cluster, resulting in $U^{meta}$ comprising one large segment and 5 - 6 smaller segments. Consequently, it appears that GMM struggles to discern nuanced differences between many segments compared to other methods, effectively smoothing out variations from different segments into a single Gaussian curve.

## C    Evaluation Settings

In our evaluation, we employ two settings. In the first setting, an item can be deemed relevant if it has already been rated or liked by the user during training. In the second setting, an item can be considered relevant only if it has not been rated or known to be liked by the user during training, and its attributes align with the user's preferences. The "Evaluation Setting for Training Items" column in the results specifies the chosen Evaluation Setting. If the column indicates "Relevant", it implies that an item recommended and known to be liked by the user can be considered relevant. If the column indicates "Irrelevant", relevance is determined only for items that were not rated by the user during the training phase.

## D    Performance of Different Segmentation Steps across Multiple Number of User Segments

For *citeulike-a* dataset in Table  4, a significant variability in model performance is observed as the number of user segments increases. Analyzing the K-Means segmentation step initially, our model demonstrates superior performance when the model size is reduced to 10%, representing 5,551/10 users. However, as the number of segments continues to increase, the results become more nuanced.

Notably, our model consistently outperforms the baseline in terms of the Mean Reciprocal Rank (MRR) metric and precision, regardless of the number of user segments. The superiority of our model over the baseline is sustained for the Recall metric only when the user segments are reduced to 10%, i.e., 5,551/10. Subsequently, as the number of segments increases, our model's Recall metric performance deteriorates. Moreover, when the model is reduced to 25% of its original size, representing 5,551/4 users, our model surpasses the baseline across all metrics, with the exception of Recall.

Considering Hierarchical K-Means, when we look at the Evaluation Setting where recommending items already seen by the user is considered to be relevant, it exhibits superior performance over the baseline across all metrics when a large number of user segments are used. Remarkably, it also performs well when the number of user segments is extremely limited, surpassing the baseline for all metrics except recall. However, when we consider the items known to or already liked by the user as irrelevant, the trend is quite

Table 4: Different Segmentation Steps across Multiple Number of User Segments for citeulike-a

| #User Segments | Method | Evaluation Setting for Training Items | % Change in Metric from Baseline | | | | |
|---|---|---|---|---|---|---|---|
| | | | Recall@20 | Precision@20 | nDCG@20 | MRR@20 | RR@1 |
| 5551/10 | KMeans | Relevant | +0.62 | +2.04 | +0.98 | +1.95 | +1.10 |
| | | Irrelevant | +0.35 | +1.65 | +0.46 | +1.20 | +0.54 |
| | HK-Means | Relevant | -3.38 | +1.11 | +0.78 | +1.22 | +0.48 |
| | | Irrelevant | -8.86 | -2.94 | -3.01 | -3.71 | -3.90 |
| | GMM | Relevant | -10.21 | -2.33 | -0.15 | -2.72 | -5.86 |
| | | Irrelevant | -8.45 | -4.14 | -1.78 | -5.99 | -4.07 |
| 5551/5 | KMeans | Relevant | -2.06 | +1.20 | -1.07 | +0.52 | -0.65 |
| | | Irrelevant | +0.35 | +1.55 | -1.26 | -0.47 | -1.77 |
| | HK-Means | Relevant | -3.25 | +0.20 | -0.45 | +0.33 | +0.15 |
| | | Irrelevant | -2.76 | +0.90 | -0.64 | +0.63 | -0.25 |
| | GMM | Relevant | -10.94 | -3.51 | -2.11 | -3.87 | -2.62 |
| | | Irrelevant | -5.28 | -2.16 | -0.70 | -2.61 | -0.96 |
| 5551/4 | KMeans | Relevant | -0.88 | +2.10 | +0.27 | +1.98 | +1.58 |
| | | Irrelevant | +0.34 | +1.72 | +0.43 | +0.10 | +1.17 |
| | HK-Means | Relevant | -4.65 | -0.72 | -0.94 | -0.46 | -0.04 |
| | | Irrelevant | -2.74 | +0.71 | -0.03 | +0.55 | +0.09 |
| | GMM | Relevant | -12.13 | -5.09 | -2.89 | -5.75 | -3.95 |
| | | Irrelevant | -2.82 | -2.17 | -0.13 | -1.99 | +0.34 |
| 5551/3 | KMeans | Relevant | -0.86 | +1.70 | -0.31 | +1.40 | +0.27 |
| | | Irrelevant | +0.34 | +1.72 | -0.09 | +0.98 | -1.05 |
| | HK-Means | Relevant | -3.19 | -0.07 | -0.93 | -0.38 | -1.11 |
| | | Irrelevant | -2.03 | +0.75 | +0.37 | +0.92 | +0.32 |
| | GMM | Relevant | -13.6 | -6.60 | -3.96 | -7.31 | -5.15 |
| | | Irrelevant | -2.07 | -1.94 | -0.70 | -2.38 | -1.21 |
| 5551/2 | KMeans | Relevant | -1.69 | +1.00 | -0.66 | +0.45 | -0.62 |
| | | Irrelevant | +1.72 | +2.03 | +0.34 | +1.70 | +0.24 |
| | HK-Means | Relevant | +0.03 | +1.62 | +0.03 | +1.16 | +0.61 |
| | | Irrelevant | +1.38 | +2.19 | +0.28 | +1.69 | +1.39 |
| | GMM | Relevant | -13.83 | -6.14 | -4.38 | -6.41 | -5.86 |
| | | Irrelevant | 0.00 | -0.10 | +0.65 | +0.30 | +1.05 |

the opposite. Also, for intermediate cases, particularly when the model size is reduced to 25% and 33.33%, Hierarchical K-Means exhibits suboptimal performance compared to the baseline.

This disparity in performance may be attributed to the inherent characteristics of Hierarchical K-Means clustering. Unlike K-Means, which maintains a fixed number of S segments at each epoch for $U^{meta}$, Hierarchical K-Means gradually reduces the number of segments to S. Consequently, for scenarios such as the 5,551/10 case, where it must converge to a significantly small S value, it may undergo more epochs and, as a result, better train the model. Conversely, its strong performance at 5,551/2 may be attributed to minimal deviation from the baseline, as fewer data points merge into a segment compared to other S values.

Referring to the *citeulike-t* dataset in Table 5, we observe that K-Means, as a segmentation step, exhibits its optimal performance when the model is reduced to 10% of its original size, equivalent to 7947/10 user segments. For other configurations, it demonstrates slightly lower performance compared to the baseline. In contrast, Hierarchical K-Means as a segmentation step follows a different trend than the results for *citeulike-a*. It performs less effectively when the model size is either very small or reduced to 50% of its original size. However, it excels in nearly all metrics, except for recall, in the intermediate scenarios, particularly when

Table 5: Different Segmentation Steps across Multiple Number of User Segments for citeulike-t

| #User Segments | Method | Evaluation Setting for Training Items | % Change in Metric from Baseline | | | | |
| | | | Recall@20 | Precision@20 | nDCG@20 | MRR@20 | RR@1 |
|---|---|---|---|---|---|---|---|
| 7947/10 | KMeans | Relevant | +0.89 | +2.51 | +2.09 | +1.95 | +1.03 |
| | | Irrelevant | -1.12 | +3.66 | +2.59 | +3.55 | +2.98 |
| | HK-Means | Relevant | -9.52 | -1.61 | -1.30 | -1.48 | -1.91 |
| | | Irrelevant | -7.78 | -0.76 | -1.18 | -0.98 | -1.43 |
| | GMM | Relevant | -11.36 | -12.64 | -5.71 | -11.93 | -7.31 |
| | | Irrelevant | -18.75 | -16.05 | -10.66 | -17.79 | -13.31 |
| 7947/5 | KMeans | Relevant | -1.74 | -0.28 | -1.24 | -0.99 | -1.32 |
| | | Irrelevant | -2.22 | +1.48 | -0.21 | +0.88 | -0.22 |
| | HK-Means | Relevant | -4.67 | -0.05 | 0.00 | -0.34 | -0.49 |
| | | Irrelevant | -2.22 | +0.76 | +0.08 | +0.13 | -0.18 |
| | GMM | Relevant | -10.97 | -12.84 | -8.55 | -13.27 | -9.97 |
| | | Irrelevant | -18.07 | -15.70 | -10.32 | -17.05 | -14.18 |
| 7947/4 | KMeans | Relevant | -1.44 | +0.48 | -0.37 | +0.26 | +0.50 |
| | | Irrelevant | -1.11 | +2.60 | +0.17 | +1.49 | +0.13 |
| | HK-Means | Relevant | -0.93 | +2.30 | +0.76 | +1.93 | +0.57 |
| | | Irrelevant | 0.00 | +2.97 | +1.02 | +2.43 | +1.11 |
| | GMM | Relevant | -10.53 | -10.58 | -5.40 | -10.59 | -7.33 |
| | | Irrelevant | -16.67 | -14.18 | -9.33 | -15.39 | -11.17 |
| 7947/3 | KMeans | Relevant | -1.70 | +0.11 | -0.70 | -0.07 | -0.69 |
| | | Irrelevant | -3.33 | +2.56 | -0.59 | +1.30 | -0.62 |
| | HK-Means | Relevant | -0.94 | +1.71 | +1.05 | +1.66 | +1.48 |
| | | Irrelevant | +2.30 | +2.95 | +1.36 | +2.52 | +2.40 |
| | GMM | Relevant | -19.19 | -15.00 | -8.77 | -14.21 | -9.82 |
| | | Irrelevant | -16.28 | -13.64 | -9.49 | -14.16 | -11.47 |
| 7947/2 | KMeans | Relevant | -1.11 | -0.02 | -0.59 | +0.10 | -0.46 |
| | | Irrelevant | +2.30 | +4.24 | +0.67 | +2.83 | +1.02 |
| | HK-Means | Relevant | -2.45 | -20.29 | -13.03 | -30.71 | -30.75 |
| | | Irrelevant | -18.21 | -31.09 | -13.90 | -31.09 | -31.28 |
| | GMM | Relevant | -20.79 | -16.37 | -9.53 | -15.19 | -10.35 |
| | | Irrelevant | -17.24 | -13.11 | -8.42 | -13.19 | -9.30 |

the model size is reduced to 25% and 33%. In the amazon dataset, there is no distinct pattern observed in the variation of the number of user segments. However, each type of segmentation step demonstrates excellent performance at a specific number of user segments. For instance, K-Means as a segmentation step exhibits improvements for all metrics at 26165/4, while a similar trend is noticed for Hierarchical K-Means and GMM at 26165/5.

GMM as a segmentation step unfortunately fails to perform better than the baseline for both the Citeulike datasets and also in many cases for the Amazon dataset which is surprising since K-Means as a segmentation step works so well. On further analysis it was discovered that K-Means seems to create few extremely large user segments and then multiple extremely small ones, reducing the granularity in the recommendations for multiple users.

Community Detection as the segmentation step is not included in this analysis because in that method we do not fix a value of the number of user segments and we allow the communities detected to be the segments. More analysis on this segmentation approach is given in Appendix F.2.

Table 6: Different Segmentation Steps across Multiple Number of User Segments for amazon

| #User Segments | Method | Evaluation Setting for Training Items | % Change in Metric from Baseline | | | | |
| --- | --- | --- | --- | --- | --- | --- | --- |
| | | | Recall@20 | Precision@20 | nDCG@20 | MRR@20 | RR@1 |
| | KMeans | Relevant | +9.33 | +8.38 | -2.64 | 4.59 | -0.68 |
| | | Irrelevant | +9.37 | +11.22 | +6.16 | +13.58 | +10.70 |
| 26165/10 | HK-Means | Relevant | +3.73 | +3.74 | -1.94 | +0.14 | -4.02 |
| | | Irrelevant | -5.92 | -6.02 | -11.64 | -10.26 | -13.77 |
| | GMM | Relevant | -2.08 | +2.95 | +0.93 | -0.47 | -1.45 |
| | | Irrelevant | +275.00 | +19.42 | +13.89 | +61.25 | -7.59 |
| | KMeans | Relevant | -12.73 | -3.21 | +4.49 | -11.21 | -4.14 |
| | | Irrelevant | -8.33 | +2.64 | +8.37 | -6.85 | -0.17 |
| 26165/5 | HK-Means | Relevant | +10.89 | +8.76 | +0.91 | +3.67 | -1.60 |
| | | Irrelevant | +7.22 | +9.33 | +1.89 | +7.03 | +2.60 |
| | GMM | Relevant | +8.73 | +9.10 | +4.36 | +7.21 | +7.35 |
| | | Irrelevant | +61.76 | +29.24 | +20.08 | +7.93 | +24.33 |
| | KMeans | Relevant | +4.72 | +7.38 | +4.07 | +5.70 | +2.49 |
| | | Irrelevant | +6.18 | +8.58 | +4.82 | +9.01 | +6.91 |
| 26165/4 | HK-Means | Relevant | +8.65 | +9.01 | -0.06 | +4.03 | -3.02 |
| | | Irrelevant | +6.18 | +9.14 | +1.33 | +5.16 | +2.29 |
| | GMM | Relevant | +0.88 | +9.00 | +2.25 | +4.68 | -1.03 |
| | | Irrelevant | -17.18 | -19.83 | -5.82 | -16.18 | -7.23 |
| | KMeans | Relevant | -19.61 | -20.98 | -5.84 | -30.91 | -23.66 |
| | | Irrelevant | -38.95 | -16.03 | -10.34 | -25.93 | -19.29 |
| 26165/3 | HK-Means | Relevant | +11.65 | +10.94 | -0.38 | +5.72 | -1.90 |
| | | Irrelevant | +8.33 | +9.09 | +0.49 | +5.41 | -0.98 |
| | GMM | Relevant | +2.80 | +1.46 | -4.27 | -1.90 | -5.83 |
| | | Irrelevant | +28.57 | +11.68 | +42.06 | +42.04 | +63.43 |
| | KMeans | Relevant | +8.57 | +7.76 | +1.96 | +4.07 | +0.16 |
| | | Irrelevant | +7.04 | +11.23 | +3.39 | +9.06 | +6.02 |
| 26165/2 | HK-Means | Relevant | -4.81 | -5.92 | -15.51 | -26.03 | -20.31 |
| | | Irrelevant | +6.31 | +7.70 | +0.35 | +4.66 | +1.52 |
| | GMM | Relevant | +0.73 | +7.64 | +0.78 | +2.56 | -1.32 |
| | | Irrelevant | -37.50 | -7.52 | -18.53 | -28.99 | -42.50 |

For the analysis with different number of user segments, we fix the number of relevant item clusters for fair evaluation for every dataset. Next we will see how changing the number of relevant item groups changes the values of the metric of our model.

## E  Performance of Different Segmentation Steps across Multiple Number of Relevant Item Clusters

Remarkably, the performance of both K-Means and Hierarchical K-Means as segmentation steps displays minimal variance in response to changes in the number of relevant item clusters.

The substantial percentage increase in performance for Community Detection as the segmentation step for *citeulike-a* may initially seem surprising. However, this outcome is consistent with the fact that this method performs poorly even in comparison to the baseline. As a result, even a significant improvement, while notable in percentage terms, still falls short of matching the baseline performance and the performance of our method when K-Means or Hierarchical K-Means segmentation steps are employed. This high percentage in increase is not seen when the items with which the model was trained on is not considered relevant. In this case, the values look rather normal. This indicates that the extremely high increase in performance is due to the recommended items being the same as the items with which the model was trained on.

However, when we look at Community Detection as the segmentation step for *citeulike-t* we see a performance where the change in percentage of the value of any of the metrics is either significantly higher or significantly lower and we are not able to discern a pattern from it. Hence it can be said that Community Detection performs erratically within and across datasets. Further insights into this matter are provided in Appendix F.2.

GMM consistently demonstrates inferior performance compared to the baseline across various configurations for the CiteULike datasets. However, a closer examination of the Amazon results in Table 9 reveals exceptionally large and unrealistic values for GMM, especially when the training data is irrelevant. This anomaly may stem from GMM recommending items that do not belong to other users in the same segment, as discussed in Section 4.3. Consequently, the relevant items that contribute somewhat normal scores when the training data is considered relevant are the only types of items recommended by GMM. Essentially, GMM derives its metric values from items that are already rated by users, highlighting a significant drawback of using GMM as a segmentation step. All of these results are computed when the number of user segments are 10% of the total number of users.

## F  Community Detection as Segmentation Step

Among the other segmentation step techniques explored was community detection that didn't lead to good results.

### F.1  Community Detection Algorithms

An alternative method for user segmentation involves utilizing graph community detection, and we employ the Louvain algorithm for this purpose, as introduced by Blondel et al. (2008). In our approach, instead of running just one iteration of the Louvain algorithm per epoch, we execute the complete process for the matrix U. Initially, each user is considered as a distinct segment, which subsequently consolidates into various clusters during subsequent iterations.

If the CDL with segmentation process fails to achieve convergence, even after an extensive number of epochs and surpasses a specific threshold, we exit the CDL process and solely execute the segmentation step iteratively until convergence is attained.

Table 7: Different Segmentation Steps across Multiple Number of Relevant Item Clusters for citeulike-a

| #Item Clusters | Method | Evaluation Setting for Training Items | % Change in Metric from Baseline | | | | |
|---|---|---|---|---|---|---|---|
| | | | Recall@20 | Precision@20 | nDCG@20 | MRR@20 | RR@1 |
| 16980/100 | KMeans | Relevant | +1.21 | +1.55 | +0.34 | +3.78 | +0.88 |
| | | Irrelevant | +0.00 | +1.62 | +0.36 | +1.32 | +0.28 |
| | HK-Means | Relevant | -1.20 | +1.76 | +0.42 | +1.62 | +0.57 |
| | | Irrelevant | -7.21 | -2.29 | -2.71 | -2.90 | -2.79 |
| | GMM | Relevant | -4.45 | -1.28 | +0.30 | -1.49 | -0.14 |
| | | Irrelevant | -6.54 | -2.17 | -0.77 | -3.29 | -1.88 |
| | Community Detection | Relevant | +1135.00 | +553.70 | +223.37 | +479.75 | +329.56 |
| | | Irrelevant | -5.26 | +15.90 | -1.54 | +2.57 | -14.28 |
| 16980/50 | KMeans | Relevant | +0.62 | +2.04 | +0.98 | +1.95 | +1.10 |
| | | Irrelevant | +0.35 | +1.65 | +0.46 | +1.20 | +0.54 |
| | HK-Means | Relevant | -3.38 | +1.11 | -0.45 | +1.22 | +0.48 |
| | | Irrelevant | -8.86 | -2.94 | -3.01 | -3.71 | -3.90 |
| | GMM | Relevant | -10.21 | -2.33 | -0.15 | -2.72 | -5.86 |
| | | Irrelevant | -8.45 | -4.14 | -1.78 | -5.99 | -4.07 |
| | Community Detection | Relevant | +1282.61 | +638.49 | +282.12 | +685.91 | +443.79 |
| | | Irrelevant | +4.54 | +43.14 | +28.41 | +52.09 | +45.04 |
| 16980/25 | KMeans | Relevant | 0.00 | +2.02 | +1.07 | +2.22 | +1.36 |
| | | Irrelevant | +0.28 | +2.66 | +1.38 | +2.79 | +3.10 |
| | HK-Means | Relevant | -4.52 | +1.33 | +0.58 | +1.14 | +0.75 |
| | | Irrelevant | -9.97 | -4.30 | -4.92 | -5.62 | -5.35 |
| | GMM | Relevant | -8.73 | -4.16 | -1.00 | -4.14 | -9.17 |
| | | Irrelevant | -12.17 | -6.79 | -3.22 | -8.17 | -4.88 |
| | Community Detection | Relevant | +1350.00 | +39.74 | +330.48 | +744.30 | +475.53 |
| | | Irrelevant | -12.00 | +24.37 | +13.29 | +31.01 | +22.21 |
| 16980/10 | KMeans | Relevant | -0.60 | +2.77 | +0.08 | +1.26 | -0.74 |
| | | Irrelevant | +1.20 | +4.40 | +1.53 | +3.80 | +1.52 |
| | HK-Means | Relevant | -4.95 | +1.06 | +0.76 | +1.15 | +1.66 |
| | | Irrelevant | -13.06 | -3.78 | -4.15 | -4.82 | -4.61 |
| | GMM | Relevant | -10.63 | -6.08 | -1.67 | -5.89 | -0.50 |
| | | Irrelevant | -12.25 | -7.00 | -2.65 | -7.85 | -3.05 |
| | Community Detection | Relevant | +1446.85 | +807.67 | +380.30 | +720.83 | +490.61 |
| | | Irrelevant | -3.85 | +41.90 | +0.05 | +0.75 | +0.75 |
| 16980/5 | KMeans | Relevant | 0.00 | +4.77 | +2.36 | +5.59 | +3.79 |
| | | Irrelevant | +0.63 | +4.09 | +3.62 | +7.07 | +7.86 |
| | HK-Means | Relevant | -4.06 | +2.92 | +1.90 | +4.21 | +3.06 |
| | | Irrelevant | -13.23 | -2.44 | -3.30 | -3.09 | -2.53 |
| | GMM | Relevant | -9.17 | -5.13 | -1.40 | -4.82 | -1.43 |
| | | Irrelevant | -11.16 | -9.37 | -5.29 | -13.01 | -9.98 |
| | Community Detection | Relevant | +1550.00 | +894.64 | +411.07 | +889.29 | +553.50 |
| | | Irrelvant | +3.45 | +57.52 | +52.43 | +46.74 | +39.65 |

Table 8: Different Segmentation Steps across Multiple Number of Relevant Item Clusters for citeulike-t

| # Item Clusters | Method | Evaluation Setting for Training Items | % Change in Metric from Baseline | | | | |
| --- | --- | --- | --- | --- | --- | --- | --- |
| | | | Recall@20 | Precision@20 | nDCG@20 | MRR@20 | RR@1 |
| 25975/100 | KMeans | Relevant | +0.39 | +1.22 | -0.03 | +0.84 | +0.04 |
| | | Irrelevant | +1.47 | +3.38 | +1.63 | +2.63 | +1.94 |
| | HK-Means | Relevant | -5.33 | -0.47 | -0.24 | -0.72 | -0.08 |
| | | Irrelevant | -5.88 | -0.45 | -1.29 | -1.64 | -1.87 |
| | GMM | Relevant | -7.46 | -6.39 | -3.69 | -5.74 | -4.23 |
| | | Irrelevant | -9.84 | -11.39 | -8.36 | -11.33 | -10.42 |
| | Community Detection | Relevant | -8.69 | -10.64 | -10.67 | -9.95 | -13.61 |
| | | Irrelevant | +20.00 | +40.58 | +11.92 | +29.07 | +17.40 |
| 25975/50 | KMeans | Relevant | +0.89 | +2.51 | +2.09 | +1.95 | +1.03 |
| | | Irrelevant | -1.12 | +3.66 | +2.59 | +3.55 | +2.98 |
| | HK-Means | Relevant | -9.52 | -1.61 | -1.30 | -1.48 | -1.91 |
| | | Irrelevant | -7.78 | -0.76 | -1.18 | -0.98 | -1.43 |
| | GMM | Relevant | -11.36 | -12.64 | -5.71 | -11.93 | -7.31 |
| | | Irrelevant | -18.75 | -16.05 | -10.66 | -17.79 | -13.31 |
| | Community Detection | Relevant | +0.82 | -0.61 | +3.98 | +7.65 | +4.76 |
| | | Irrelevant | 0.00 | +41.00 | +25.42 | +26.42 | +24.06 |
| 25975/25 | KMeans | Relevant | -0.07 | +0.74 | +0.49 | +1.45 | +1.35 |
| | | Irrelevant | 0.00 | +3.73 | +3.44 | +4.61 | +4.33 |
| | HK-Means | Relevant | -7.57 | -0.87 | -0.63 | +0.16 | +0.16 |
| | | Irrelevant | -3.64 | +0.96 | +0.05 | +0.74 | +0.14 |
| | GMM | Relevant | -16.36 | -10.88 | -6.37 | -12.24 | -10.57 |
| | | Irrelevant | -9.37 | -7.79 | -4.82 | -7.11 | -6.53 |
| | Community Detection | Relevant | 0.00 | -6.82 | -4.92 | +4.54 | +5.43 |
| | | Irrelevant | 0.00 | +38.57 | +33.07 | +36.46 | +22.56 |
| 25975/10 | KMeans | Relevant | -1.12 | +0.38 | -0.06 | -0.18 | -0.04 |
| | | Irrelevant | +2.36 | +6.39 | +4.24 | +7.24 | +5.71 |
| | HK-Means | Relevant | -7.59 | -0.27 | -1.24 | -1.16 | -1.60 |
| | | Irrelevant | -2.36 | +2.62 | +0.35 | +2.00 | +0.66 |
| | GMM | Relevant | +1.74 | -1.11 | -4.64 | -10.08 | -6.57 |
| | | Irrelevant | -11.61 | -11.41 | -6.19 | -9.04 | -6.97 |
| | Community Detection | Relevant | -8.33 | -9.73 | -9.56 | +1.34 | -9.26 |
| | | Irrelevant | 0.00 | +43.53 | +51.50 | +43.13 | +35.52 |
| 25975/5 | KMeans | Relevant | -0.66 | +2.76 | -0.03 | +0.84 | +0.47 |
| | | Irrelevant | +1.97 | +7.03 | +4.70 | +8.30 | +7.33 |
| | HK-Means | Relevant | -6.52 | +0.89 | -0.13 | +1.93 | +0.66 |
| | | Irrelevant | -3.97 | +4.53 | +2.02 | +4.72 | +2.79 |
| | GMM | Relevant | -16.23 | -5.18 | -4.46 | -5.38 | -6.81 |
| | | Irrelevant | -10.32 | -5.11 | -2.36 | -3.24 | -0.17 |
| | Community Detection | Relevant | 0.00 | -16.11 | -5.21 | -4.38 | +5.55 |
| | | Irrelevant | -21.43 | +3.45 | +10.43 | -1.18 | -4.12 |

Table 9: Different Segmentation Steps across Multiple Number of Relevant Item Clusters for amazon

| # Item Clusters | Method | Evaluation Setting for Training Items | % Change in Metric from Baseline | | | | |
|---|---|---|---|---|---|---|---|
| | | | Recall@20 | Precision@20 | nDCG@20 | MRR@20 | RR@1 |
| 31245/100 | KMeans | Relevant | -6.25 | -8.35 | +6.10 | -8.33 | -0.81 |
| | | Irrelevant | +8.20 | +8.52 | +2.42 | +7.71 | +3.11 |
| | HK-Means | Relevant | +3.03 | +2.66 | -3.31 | +0.08 | -4.11 |
| | | Irrelevant | +0.69 | -1.71 | -6.73 | -4.20 | -9.06 |
| | GMM | Relevant | -3.03 | +1.07 | -0.86 | -0.14 | -1.73 |
| | | Irrelevant | -25.86 | -16.94 | +5.84 | -12.38 | +4.22 |
| 31245/50 | KMeans | Relevant | +7.32 | +9.15 | +0.48 | +6.46 | +0.22 |
| | | Irrelevant | +7.92 | +6.86 | -1.83 | +1.92 | -4.43 |
| | HK-Means | Relevant | -0.96 | -2.89 | -9.07 | -6.77 | -11.75 |
| | | Irrelevant | -3.29 | -2.50 | -8.70 | -6.47 | -11.31 |
| | GMM | Relevant | -5.85 | -0.30 | -2.83 | -3.22 | -5.85 |
| | | Irrelevant | +2.17 | +24.13 | +27.98 | +29.69 | +36.35 |
| 31245/25 | KMeans | Relevant | +9.33 | +8.38 | -2.64 | 4.59 | -0.68 |
| | | Irrelevant | +9.37 | +11.22 | +6.16 | +13.58 | +10.70 |
| | HK-Means | Relevant | +3.73 | +3.74 | -1.94 | +0.14 | -4.02 |
| | | Irrelevant | -5.92 | -6.02 | -11.64 | -10.26 | -13.77 |
| | GMM | Relevant | -2.08 | +2.95 | +0.93 | -0.47 | -1.45 |
| | | Irrelevant | +275.00 | +19.42 | +13.89 | +61.25 | -7.59 |
| 31245/10 | KMeans | Relevant | -1.16 | +1.18 | -5.17 | -3.65 | -5.93 |
| | | Irrelevant | +3.50 | +2.64 | -5.22 | -3.33 | -8.80 |
| | HK-Means | Relevant | -8.08 | -7.43 | -14.50 | -13.37 | -16.73 |
| | | Irrelevant | -7.56 | -5.87 | -13.93 | -11.88 | -15.59 |
| | GMM | Relevant | -9.17 | -2.22 | -8.47 | -4.43 | -9.02 |
| | | Irrelevant | -18.18 | -6.96 | +15.15 | -3.89 | +8.41 |
| 31245/5 | KMeans | Relevant | -2.18 | -0.32 | -7.64 | -7.10 | -10.07 |
| | | Irrelevant | -24.79 | -23.55 | -14.51 | -24.61 | -20.02 |
| | HK-Means | Relevant | -8.67 | -8.82 | -16.31 | -15.98 | -18.44 |
| | | Irrelevant | +8.72 | +11.42 | -0.73 | +5.15 | -1.52 |
| | GMM | Relevant | -8.53 | -1.87 | -6.14 | -12.43 | -12.81 |
| | | Irrelevant | -24.00 | -7.82 | -20.18 | -20.55 | -20.63 |

Table 10: Community Detection Results for citeulike-a and citeulike-t

| Evaluation Setting for training items | Experimental Setting | Recall@20 | Precision@20 | nDCG@20 | MRR@20 | RR@1 |
|---|---|---|---|---|---|---|
| **citeulike-a:** | | | | | | |
| | Proposed Model | 0.0318 | 0.5487 | 0.6691 | 2.008 | 0.6346 |
| Relevant | Baseline Model | 0.0023 | 0.0743 | 0.1751 | 0.2555 | 0.1167 |
| | Change(%) | +1282.61 | +638.49 | +282.12 | +685.91 | +443.79 |
| | Proposed Model | 0.0023 | 0.0710 | 0.2002 | 0.2943 | 0.1565 |
| Irrelevant | Baseline Model | 0.0022 | 0.0496 | 0.1559 | 0.1935 | 0.1079 |
| | Change(%) | +4.54 | +43.14 | +28.41 | +52.09 | +45.04 |
| **citeulike-t:** | | | | | | |
| | Proposed Model | 0.0123 | 0.0811 | 0.2483 | 0.4739 | 0.1540 |
| Relevant | Baseline Model | 0.0122 | 0.0815 | 0.2388 | 0.4402 | 0.1470 |
| | Change(%) | +0.82 | -0.49 | +3.98 | +7.65 | +4.76 |
| | Proposed Model | 0.0008 | 0.0282 | 0.0592 | 0.1067 | 0.0495 |
| Irrelevant | Baseline Model | 0.0008 | 0.0200 | 0.0472 | 0.0844 | 0.0399 |
| | Change(%) | 0.00 | +41.00 | +25.42 | +26.42 | +24.06 |

### F.2 Results

Table 10 highlights a substantial percentage increase in most metric values compared to the baseline, notably higher than observed with other segmentation methods especially for *citeulike-a*. However, these results must be discounted due to the significant degradation in baseline performance caused by this algorithm. However, when comparing these results with those of the baseline and our method for the *citeulike-a* dataset with K-Means, as presented in Table 1, it becomes evident that the results in Table 1 exhibit a slight improvement over those obtained with Community Detection as the segmentation step for the proposed model.

However, in the case of *citeulike-t* results, while the percentage increase in metric values is on par with other segmentation step methods, a closer look at the absolute metric values reveals significantly lower precision, nDCG, and RR@1 scores compared to those achieved with K-Means as the segmentation step in Table 1. Notably, MRR@20 demonstrates superior performance, but it's crucial to consider that this is just one metric. The recall values remain comparable.

When employed for the Amazon dataset, Community Detection as a segmentation step causes the training to terminate before completing a single epoch due to excessive memory usage. This indicates that Community Detection is not a highly scalable segmentation method for very large datasets.

Considering the numerous drawbacks outlined for Community Detection as a segmentation step, it can be concluded that it is not a suitable method for user segmentation in any scenario. This may be because there is no way of specifying the size of user segments in this method as it automatically tries to detect that and fails to do so.

## G  User Cluster Examples

Table 11 shows the preferences of each of the users that belong to the segment 187. These preferences are the data with which the model has been trained and it has grouped these users together into segment 187. Table 12 compares the recommendations given to all the users of segment 187 by both our model and the baseline.

Table 11: **Results with citeulike-a** For segment: 187 Users: 1092, 2430, 5004 - These are the titles that each of the users have rated i.e. the titles the model is trained on for these users.

| Titles Trained |
| --- |
| **User 1092:** |
| (1) Exploring social dynamics in online media sharing |
| (2) Understanding the Characteristics of Internet Short Video Sharing: YouTube as a Case Study |
| (3) Measurement and analysis of online social networks |
| (4) Publicly Private and Privately Public: Social Networking on YouTube |
| (5) I tube, you tube, everybody tubes: analyzing the world's largest user generated content video system |
| (6) Robust dynamic classes revealed by measuring the response function of a social system |
| (7) Predicting the popularity of online content |
| (8) Statistics and Social Network of YouTube Videos |
| (9) YouTube: Online Video and Participatory Culture |
| (10) Users of the world, unite! The challenges and opportunities of Social Media |
| **User 2430:** |
| (1) Maximizing the Spread of Influence through a Social Network |
| (2) Analysis of recommendation algorithms for e-commerce |
| (3) The Dynamics of Viral Marketing |
| (4) Linear prediction: A tutorial review |
| (5) Link prediction approach to collaborative filtering |
| (6) Understanding the Characteristics of Internet Short Video Sharing: YouTube as a Case Study |
| (7) Content-Based Recommendation Systems |
| (8) Publicly Private and Privately Public: Social Networking on YouTube |
| (9)Youtube traffic characterization: a view from the edge |
| (10) I tube, you tube, everybody tubes: analyzing the world's largest user generated content video system |
| **User 5004:** |
| (1) Tagging, Folksonomy & Co - Renaissance of Manual Indexing? |
| (2) Understanding the Characteristics of Internet Short Video Sharing: YouTube as a Case Study |
| (3) Tagging Video: Conventions and Strategies of the YouTube Community |
| (4) Publicly Private and Privately Public: Social Networking on YouTube |
| (5) Youtube traffic characterization: a view from the edge |
| (6) I tube, you tube, everybody tubes: analyzing the world's largest user generated content video system |
| (7) Video suggestion and discovery for youtube: taking random walks through the view graph |
| (8) Predicting the popularity of online content |
| (9) Statistics and Social Network of YouTube Videos |
| (10) YouTube: Online Video and Participatory Culture |

Table 12: **Results with citeulike-a** For segment: 187 - Model recommendations for all users in Table 11

Proposed Model

(1) I tube, you tube, everybody tubes: analyzing the world's largest user generated content video system
(2) Understanding the Characteristics of Internet Short Video Sharing: YouTube as a Case Study
(3) Exploring social dynamics in online media sharing
(4) Publicly Private and Privately Public: Social Networking on YouTube
(5) Video suggestion and discovery for youtube: taking random walks through the view graph
(6) Teens and Social Media
(7) Statistics and Social Network of YouTube Videos
(8) Project massive: a study of online gaming communities
(9) Predicting the popularity of online content
(10) The Role of Friends' Appearance and Behavior on Evaluations of Individuals on Facebook: Are We Known by the Company We Keep?

Baseline Model

(1) I tube, you tube, everybody tubes: analyzing the world's largest user generated content video system
(2) Understanding the Characteristics of Internet Short Video Sharing: YouTube as a Case Study
(3) Exploring social dynamics in online media sharing
(4) Statistics and Social Network of YouTube Videos
(5) Can internet video-on-demand be profitable?
(6) Exploring social dynamics in online media sharing
(7) Youtube traffic characterization: a view from the edge
(8) Predicting the popularity of online content
(9) A Measurement Study of the BitTorrent Peer-to-Peer File-Sharing System
(10) Measurement, modeling, and analysis of a peer-to-peer file-sharing workload

Table 13: Metric values for User Segmentation when Outliers are added

| Experimental Setting | Recall@20 | Precision@20 | nDCG@20 | MRR@20 | RR@1 | ROU@20 |
|---|---|---|---|---|---|---|
| **citeulike-a:** | | | | | | |
| Proposed Model | 0.0280 | 0.4980 | 0.6746 | 1.9559 | 0.6349 | 0.2724 |
| Baseline Model | 0.0286 | 0.4861 | 0.6698 | 1.9203 | 0.6348 | 0.2492 |
| Change(%) | -0.21 | +2.45 | +0.72 | +1.85 | +0.01 | +9.31 |

## H   Robustness to Outliers

H In order to test the model's robustness to tapping the user's preferences when the user has not expressed it well, we try to analyze our model's performance when around 20% of the users have given a preference to one item that they do not like. We do this by manually intervening in the training data and changing one item preference for a random set of users. We see that our model is stable to such interventions. This is seen on comparing the plots in Figure 1 and Figure 2. The plots look very similar. The most striking similarity is seen with the yellow segment at the center for both the plots. Table 13 shows that there isn't much of a difference in the values of the metrics as compared to Table 1 and that our method is robust to outliers too.

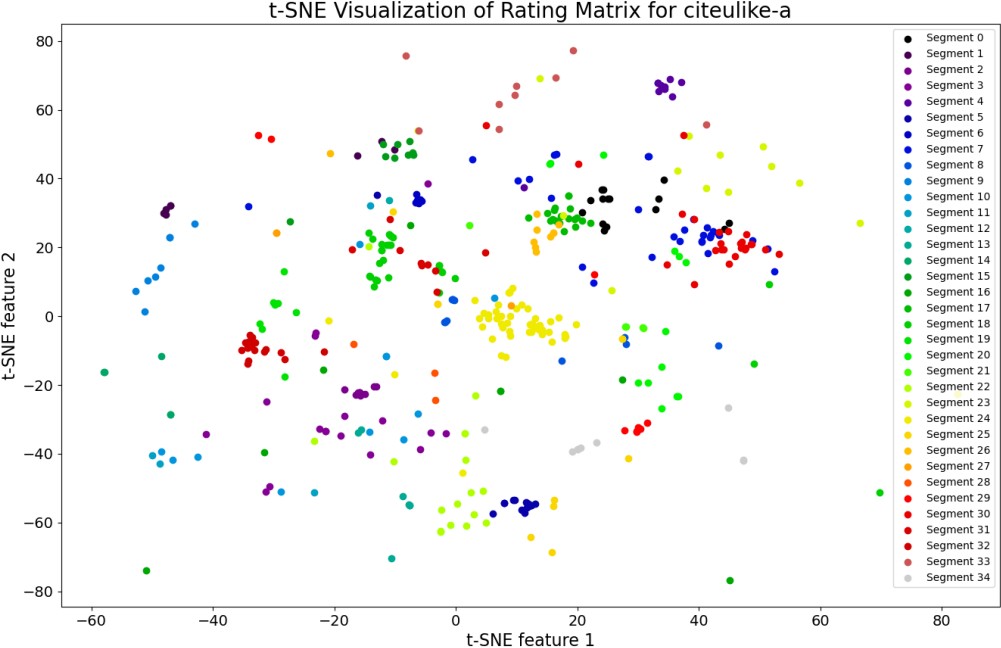

Figure 2: The ratings of users belonging to 35 segments when around 20% of the users have rated one item that they dislike

