# OpenReview forum: "User Segmentation in Recommender Systems: Problem Formulation, Algorithms, and Evaluations"
_TMLR — Rejected by TMLR_

### Review · Reviewer_MgNB · 2024-03-11

**Summary Of Contributions:**

The paper proposes to add a user segmentation/clustering step to the CDL (Wang et al. 2015) method for the recommendation system. The training objective is modified and two standard clustering methods are used (k-means and hierarchical k-means, GMM is in appendix). During recommendation time, each group will get the same recommendations. Experiments are conducted on two small, unpopular public datasets and one larger dataset from Amazon that seems to be internal. Positive results are reported upon CDL.

**Audience:**

No

**Claims And Evidence:**

No

**Requested Changes:**

Use reasonable datasets, find more convincing and recent baselines, do a literature survey and adjust your claims, fix apparent editorial errors.

**Strengths And Weaknesses:**

Strength:
- Despite the assumptions and caveats introduced, the proposed objective looks valid.
- User segmentation may be a good topic, though the reviewer is not fully convinced.

Weakness:
Unfortunately, the submission has many caveats that are not likely to be addressed in a reasonable amount of time.

- Surprisingly limited scope and weak baseline: the proposed method is only applied to CDL, which is the only baseline in experiments, and it was published in 2015. This is way below any publication standard in any venue.
- Surprisingly weak benchmark: none of the datasets used are standard.
- The reviewer is surprised that the authors seem to claim there's limited work on this direction and clearly user segmentation is an old concept in practical recommendation systems. A simple web search shows similar papers (do user segmentation, use the same set of recommendations for each group) published back to 2016 (https://www.emerald.com/insight/content/doi/10.1108/K-07-2014-0130/full/html).
- The reviewer is not convinced of the proposed approach. While the naiveness of the proposed methods may be ok, the major caveat of any clustering method - how to decide the number of segments - is not well discussed in the submission. This makes the work useless in any practical setting.
- The editorial errors and typos are easy to catch ones, so they are annoying and make the reviewer question the efforts invested. There's a typo in the first sentence for the main part (section 2) and many others.

---

> ### Author Response · Authors · 2024-04-09
> **Response to Review**
>
> Thank you for your comments. I am glad that you find the objective of this work valid. We agree that there are some weaknesses that cannot be addressed immediately. However, we can address some of the changes that have been pointed out:
>
> **1. Surprisingly limited scope and weak baseline: the proposed method is only applied to CDL, which is the only baseline in experiments, and it was published in 2015. This is way below any publication standard in any venue.**
>
> We accept that we have compared our results with only one baseline which is the CDL approach. Comparison with more baselines is future work.
>
> **2. Surprisingly weak benchmark: none of the datasets used are standard.**
>
> One of the datasets we have used, the Amazon data set is a standard one. In addition we have results on two other smaller datasets. However, we can try adding in results for another dataset which is popularly used such as the Movielens or Netflix Prize Dataset
>
> **3. The reviewer is surprised that the authors seem to claim there's limited work on this direction and clearly user segmentation is an old concept in practical recommendation systems. A simple web search shows similar papers (do user segmentation, use the same set of recommendations for each group) published back to 2016 (https://www.emerald.com/insight/content/doi/10.1108/K-07-2014-0130/full/html).**
>
> The paper mentioned by the reviewer are not really similar to our work. The metric that this paper uses is the metrics of recency, frequency and monetary associated with a user to perform the clustering. This is very different from our approach which relies on the user’s product preferences to perform the clustering. Using product preferences seems a more natural way to cluster users. In fact, that is the first metric one would think of to cluster users.
>
> **4. The reviewer is not convinced of the proposed approach. While the naiveness of the proposed methods may be ok, the major caveat of any clustering method - how to decide the number of segments - is not well discussed in the submission. This makes the work useless in any practical setting.**
>
> This is completely orthogonal to our problem and is an issue in all application areas where clustering is used, not just in recommender systems. Any recommender systems used in a commercial setting will need to be extensively tuned to improve results. Determining number of clusters would be one such parameter that would be tuned to get best results. In this context, please see also the work in recent papers such as https://scholar.google.com/citations?view_op=view_citation&hl=en&user=zX3ba1kAAAAJ&sortby=pubdate&citation_for_view=zX3ba1kAAAAJ:buQ7SEKw-1sC in which the authors discuss clustering algorithms with proven approximation ratios. It seems likely that clustering algorithms with theoretical guarantees could be leveraged which would enable us to know how far we are away from the optimal clustering which, in turn, would enable us to alter the clustering to move closer to the optimal. But, as mentioned before, this is orthogonal to our work and any clustering approach which allows the ideal number of clusters to be determined can be leveraged in our work.
>
> **5. The editorial errors and typos are easy to catch ones, so they are annoying and make the reviewer question the efforts invested. There's a typo in the first sentence for the main part (section 2) and many others.**
>
> We apologize for the editorial errors and typos and make sure that they are fixed.

---

> > ### Comment · Reviewer_MgNB · 2024-04-09
> > **Thank you**
> >
> > Thank you for the reply. The paper likely needs substantial revisions and hope the reviews can help.

---

### Review · Reviewer_k87N · 2024-03-19

**Summary Of Contributions:**

The paper considers the problem of personalized recommendation. The starting point is the Collaborative Deep Learning (CDL) algorithm that combines matrix factorization with an autoencoder to learn the item latent vectors.
The new idea here is to cluster to user latent vectors in CDL. During recommendation the individual user latent vector is replaced by the average vector of the cluster where the user belongs. The clustering is based on the user latent vectors learned by CDL. Various clustering approaches are considered, including k-means, hierarchical k-means, Gaussian Mixture Models.
The empirical evaluation uses various standard measures such as recall, precision, MRR, NDCG, but also some clustering specific ones as well. Three moderately small dataset are used, and the new algorithm is compared to a CDL baseline. The experimental results indicate that the proposed algorithm improves on the baseline for all measures considered.

**Audience:**

Yes

**Broader Impact Concerns:**

No concerns.

**Claims And Evidence:**

No

**Requested Changes:**

Given that this is a largely empirical paper, the experiments should be far more extensive, and described in more detail.

**Strengths And Weaknesses:**

The paper is relatively well structured. Nevertheless, it would be better, if the prediction model would be described first, before define the optimization problem define by equation (1)-(5).

I am not convinced by the argument for user clustering. In some sense it is a depersonalization of the personalized recommendation. There are hierarchical matrix factorization approaches where user latent vectors are constrained to be close to the latent vectors of the cluster (or the latent vectors are combined), which smooth out the noise, but still keep the recommendation personal.

The baselines used in the empirical evaluation is really limited. Even if the study limits to matrix factorization, there are a vast number of algorithms that should be considered, included hierarchical and implicit variants.

There is very little information about the data sets, and the sets are very small compared to typical data sets used even 10 years ago.

There is no description, but I assume that each data point is the result of on run. Nothing about significance, not even k-fold cross validation.

---

> ### Author Response · Authors · 2024-04-09
> **Response to Review**
>
> Thank you for your feedback. As per the requested changes, we can add in some more details to our description of the experiments. Some of the weaknesses that have been identified cannot be dealt with immediately, but we have tried to address them in the best way possible:
>
> **1. The paper is relatively well structured. Nevertheless, it would be better, if the prediction model would be described first, before define the optimization problem defined by equation (1)-(5).**
>
> We have taken this comment into consideration and now we have the prediction model described first.
>
> **2. I am not convinced by the argument for user clustering. In some sense it is a depersonalization of the personalized recommendation.**
>
> Our primary objective is to ensure user satisfaction with the recommendations presented, assessed through several metrics outlined in our study. When these metrics indicate superior outcomes for our approach compared to more personalized techniques, it suggests that our depersonalization strategy leads to more effective recommendations. Furthermore, our paper details our rationale for adopting a depersonalization approach, which includes recognizing patterns within user groups and mitigating the noise in individual preferences.
>
> **3. There are hierarchical matrix factorization approaches where user latent vectors are constrained to be close to the latent vectors of the cluster (or the latent vectors are combined), which smooth out the noise, but still keep the recommendation personal.**
>
> We acknowledge the potential of Hierarchical Matrix Factorization in addressing our goal to develop a recommendation system that considers user relationships and mitigates noise. Nevertheless, this approach does not inherently yield distinct user groups or segments that could be directly leveraged for marketing strategies, in contrast to our proposed method. To clarify this distinction, we have included a comparison between Hierarchical Matrix Factorization and our methodology in the related work section of our paper. While Hierarchical Matrix Factorization could serve as a foundational comparison point for our research, exploring its application as a baseline constitutes an avenue for future work.
>
> **4. The baselines used in the empirical evaluation is really limited. Even if the study limits to matrix factorization, there are a vast number of algorithms that should be considered, included hierarchical and implicit variants.**
>
> We accept that we have compared our results with only one baseline which is the CDL approach. Comparison with more baselines is future work.
>
> **5. There is very little information about the data sets, and the sets are very small compared to typical data sets used even 10 years ago.**
>
> We have added more details related to the datasets and how they have been obtained. We agree that the datasets are very small and we can try adding in results for another dataset which is popularly used such as the Movielens or Netflix Prize Dataset.
>
> **6. There is no description, but I assume that each data point is the result of on run. Nothing about significance, not even k-fold cross validation.**
>
> In response to the concerns raised, it is important to clarify that each data point presented in our study does not represent the outcome of a single execution but rather aggregates the results from multiple iterations, each employing varied hyperparameter settings. We meticulously selected the hyperparameters that yielded the most favorable outcomes across different sizes of user and item segments throughout these iterations. Unless explicitly stated otherwise in the provided tables, the results pertaining to any specific dataset and segmentation technique were derived using a consistent hyperparameter setup.
>
> Furthermore, due to the relatively modest size of our datasets, we opted against implementing k-fold cross-validation. We recognized that partitioning our data further into smaller subsets for cross-validation would compromise the dataset's integrity, adversely affecting the robustness of our findings.

---

> > ### Comment · Reviewer_k87N · 2024-04-10
> >
> > I appreciate the authors' effort in addressing the concerns raised by the reviewers. I hope that this will lead to a significantly improved, but at this moment the paper would not too much revision to be accepted.

---

### Review · Reviewer_n3Kb · 2024-03-25

**Summary Of Contributions:**

**Summary of Contributions:**

The paper aims to address the problem of user segmentation in recommender systems. The authors propose a method that incorporates user segmentation into the Collaborative Deep Learning (CDL) framework by adding an extra term to the objective function. This term constrains each user's latent vector to be close to its segment's latent vector. The proposed approach is evaluated using several segmentation algorithms (K-means, hierarchical K-means, GMM, and community detection) on three diverse datasets. The authors compare the performance of their method against a baseline that applies segmentation after the CDL training process. The results show improvements of 9-13% across five standard evaluation metrics on a large dataset and 1-2% improvements on two smaller datasets. The authors also introduce a custom metric, Relevance to Other Users, which improves by around 20% with their approach. Additionally, the paper demonstrates that the model size can be significantly reduced (e.g., to 10% of the original size with K-means) while maintaining strong performance. A qualitative analysis is provided to examine the recommendations made to an example user segment.

**Audience:**

Yes

**Claims And Evidence:**

No

**Requested Changes:**

Essentially, all the 8 weaknesses that I described above are a potential source of changes that the authors can make for a future submission.

**Strengths And Weaknesses:**

**Strengths and Weaknesses:**

The paper is well written and puts itself nicely in context of past work.

**Strengths:**

1). The paper tackles the problem of user segmentation in recommender systems, which is an interesting and relevant topic.

2). The authors evaluate their approach using multiple segmentation algorithms and compare performance across various hyperparameter configurations and datasets.

3). The proposed method is able to reduce model size significantly while maintaining performance.

4). The paper includes a qualitative analysis to provide insights into the recommendations made to an example user segment.

**Weaknesses:**

1). *Lack of Methodological Novelty*: The proposed method is essentially a wrapper around Collaborative Deep Learning (CDL), with an additional term added to the objective function (Equation 8). The authors do not provide any theoretical justification or optimality criteria proofs for including this term. The paper fails to demonstrate how this modification leads to a principled improvement over existing techniques. Without a solid theoretical foundation, the proposed approach appears to be an ad-hoc extension of CDL, lacking in methodological novelty.

2). *Marginally significant results*: The reported results do not show a substantial improvement in performance. As evident from Table 1, the proposed method yields improvements of only 0.35%, 1.65%, and similarly small percentages across various metrics and datasets. These improvements are likely within the margin of error and do not justify the need for the proposed approach. For a paper introducing a new method, one would expect to see more significant gains over baselines to warrant publication.

3). *Questionable Relevance of User Segmentation*: The main idea behind the proposed approach, user segmentation, raises concerns about its relevance in the current context. In the era of hyper-personalized individual-level targeting for advertisements and recommendations, the segment-level approach presented in the paper feels outdated. It resembles techniques more prevalent in the early days of the Internet, such as the 1990s, when granular user data was limited. The authors do not make a compelling case for why user segmentation is still a valuable pursuit, given the shift towards individual-level personalization in recent years.

4). *Misfit with Journal Scope*: Given the focus on user segmentation and the potential applications in marketing and business domains, this paper seems better suited for a journal in those areas. The lack of methodological novelty and the emphasis on a somewhat dated approach make it a suboptimal fit for a cutting-edge research venue such as TMLR.

5). *Insufficient Guidance on Hyperparameter Tuning*: The paper does not provide sufficient guidance on selecting the optimal number of user segments, which appears to vary significantly across datasets. A more systematic approach to tuning this crucial hyperparameter is needed.

6). *Unstable Performance of Some Segmentation Methods*: Some of the evaluated segmentation methods, such as GMM and community detection, exhibit unstable performance. The authors should discuss the limitations of these methods and provide clearer guidelines on when they are unsuitable.

7). *Limited Evaluation Metrics*: The evaluation primarily focuses on accuracy metrics, neglecting other important factors in recommender systems, such as novelty, diversity, and explainability. A more comprehensive assessment would strengthen the work.

8). *Lack of Deployment Considerations*: The paper lacks a thorough discussion of deployment considerations, such as the computational efficiency of different segmentation methods and the practicality of frequent user re-clustering. Addressing these aspects is crucial for understanding the real-world applicability of the proposed approach.

---

> ### Author Response · Authors · 2024-04-09
> **Response to Review**
>
> Thank you for your feedback. We would like to address your concerns:
>
> **1. Lack of Methodological Novelty**
>
> The paper describes empirical studies on the use of different cluster approaches to improve accuracy and size of the model in recommender systems. Theoretical justification will be part of future work.
>
> **2. Marginally Significant Results**
>
> We accept that there is only a 1-2% improvement in a lot of results in the citeulike datasets, however we have 9-13% improvement on our larger Amazon Dataset. This result is obtained at a significant reduction of the model size, that is about 10% of its original size. We can try adding in results for another dataset which is popularly used such as the Movielens or Netflix Prize Dataset.
>
> **3. Questionable Relevance of User Segmentation**
>
> The improved accuracy and reduced model size supports the relevancy of the user segmentation approach. Since users in the same segment are likely to be similar, personalized recommendation would not be affected. Across clusters, the recommendations would still be different. Also, the user segments can be obtained from the model and used for marketing purposes.
>
> **4. Insufficient Guidance on Hyperparameter Tuning**
>
> This is completely orthogonal to our problem and is an issue in all application areas where clustering is used not just in recommender systems. Any recommender systems used in a commercial setting will need to be extensively tuned to improve results. Determining number of clusters would be one such parameter that would be tuned to get best results. In this context, please see also the work in recent papers such as https://scholar.google.com/citations?view_op=view_citation&hl=en&user=zX3ba1kAAAAJ&sortby=pubdate&citation_for_view=zX3ba1kAAAAJ:buQ7SEKw-1sC in which the authors discuss clustering algorithms with proven approximation ratios. It seems likely that clustering algorithms with theoretical guarantees could be leveraged which would enable us to know how far we are away from the optimal clustering which, in turn, would enable us to alter the clustering to move closer to the optimal. But, as mentioned before, this is orthogonal to our work and any clustering approach which allows the ideal number of clusters to be determined can be leveraged in our work.
>
> **5. Unstable Performance of Some Segmentation Methods**
>
> This paper’s aim was to formulate the problem of User Segmentation and give an overview of the possible algorithms that can be used. Our analysis reveals that certain algorithms outperform others in terms of effectiveness. Consequently, to maintain transparency and comprehensiveness in reporting, we have relegated the outcomes of the less stable algorithms to the Appendix. This decision was made to ensure that the main body of our work remains focused on the more successful segmentation strategies, while still providing a complete overview of all tested algorithms.
>
> **6. Limited Evaluation Metrics**
>
> We admit that our metrics so far only focus on the accuracy of the recommendation system. We can try and include a metric for the novelty of the recommendations such as a popularity rank.
>
> **7. Lack of Deployment Considerations**
>
> We have added some information about deployment considerations in the Discussion and Conclusion section.

---

> > ### Comment · Reviewer_n3Kb · 2024-04-09
> > **Thank You.**
> >
> > I would like to thank the authors for their rebuttal. However, their rebuttal doesn't change my stance on the paper and doesn't address the weaknesses I listed.

---

### Decision · Action_Editor_EP7T · 2024-04-10

**Recommendation:** Reject

**Comment:**

The reviewers agree this submission is not technically strong enough to warrant acceptance. User segmentation is a well-studied problem and the vast majority of modern recommendation systems have moved from this paradigm into approaches heavily relying on personalisation based on user preference/latent vectors (similar to those obtained through word2vec etc.). The methods in this paper then present little interest to practitioners. To claim the conclusions in the paper generalize to modern recommendation systems, the experiments should show evidence of performance gains on a diverse set of modern algorithms and datasets.

**Audience:**

The paper does not present generalizable conclusions about the proposed approach, as that the main claims are not supported by enough evidence. The reviewers are in agreement that the paper is not technically strong enough to present interest to the audience of this journal.

**Claims And Evidence:**

The reviewers are in agreement that the evidence presented in this submission fall short of supporting the claims of the paper, and I agree with their assessment:
- the evaluation framework is insufficient as it does not cover aspects of novelty, diversity etc.
- the claim that the datasets are crafted as part of this work is not supported
- the claim that this paper introduces a novel approach to recommendation is also not supported (segmentation is not a new practice in recommender systems - see the comments of reviewer MgNB)
- the approach is only tested (and shows improvements) on a single baseline algorithm, and a very limited set of datasets, which is insufficient to claim the approach generalizes.